# Spatiotemporal Dynamics of Ecological Vulnerability and Its Influencing Factors in Shenyang City of China: Based on SRP Model

**DOI:** 10.3390/ijerph20021525

**Published:** 2023-01-14

**Authors:** Hanlong Gu, Chongyang Huan, Fengjiao Yang

**Affiliations:** College of Land and Environment, Shenyang Agricultural University, Shenyang 110866, China

**Keywords:** ecological vulnerability, principal component analysis, spatial autocorrelation, geodetector, GIS, Shenyang

## Abstract

For Shenyang, the central city of Northeast China, its municipal-level Territorial Spatial Planning is of great significance to the whole of Northeast China. Territorial Spatial Planning is an essential carrier of China’s ecological civilization construction. The demarcation of “three districts and three lines” defines the scope of ecological protection areas, which is of guiding significance to the future development of ecological civilization construction. The regional ecological vulnerability assessment can provide reference for ecological pattern planning and the demarcation of ecological red lines in “three districts and three lines”. In order to explore the spatial distribution pattern of ecological vulnerability in Shenyang, predict the development trend of ecological vulnerability in the future and guide the construction of ecological civilization in Shenyang and provide certain basis for Shenyang’s Territorial Spatial Planning and the delineation of “three districts and three lines”. This paper based on the “sensitivity-resilience-pressure” model selected 13 indexes, to evaluate the ecological vulnerability of Shenyang from 2010 to 2020. Furthermore, the spatial distribution characteristics and influencing factors of ecological vulnerability in Shenyang are summarized using spatial autocorrelation analysis and geographic detector model, and the future development trend of ecological vulnerability in Shenyang in 2025 is predicted by using CA-Markov model. The results show that: (1) In 2010, 2015 and 2020, the total area of slightly vulnerable areas in Shenyang was large, and the ecological vulnerability showed a gradually vulnerable spatial change trend from south to north and from west to east. (2) The results of geographical detectors show that normalized difference vegetation index, economic density and nighttime light intensity are the main driving factors of ecological vulnerability in Shenyang. (3) The forecast result of CA-Markov model is reliable. In 2025, the ecological vulnerability of Shenyang will be mainly light and extreme vulnerability areas, and the areas of light and extreme vulnerability areas will increase in 2025. The research results can provide some reference for the delineation of “three districts and three lines” and ecological protection in Shenyang’s Territorial Spatial Planning, and have certain significance for promoting regional sustainable development and balancing ecological protection and economic development.

## 1. Introduction

Since the 20th century, global warming has become increasingly serious, and the ecological environment has undergone dramatic changes [1]. Energy shortage, loss of species diversity, frequent extreme weather events, and increasing natural disasters such as soil erosion, desertification and polar glacier melting have greatly threatened human survival and sustainable socioeconomic development [2]. Meanwhile, the rapid expansion of population and irrational exploitation of resources have led to the decreasing recovery and self-cleaning capacity of ecosystems, and the ecological environment has become increasingly vulnerable [3]. According to the relevant data published by the United Nations Development Programme, the United Nations Environment Programme, the World Bank and the World Resources Institute, half of the world’s wetlands disappeared in the 20th century, deforestation and occupation of forest land caused the world’s forests to shrink by half, about 9% of the world’s tree species are on the verge of extinction, and more than 130,000 km^2^ of tropical forests are destroyed every year. Clements, an American ecologist, first put forward the concept of ecological vulnerability in his research on “Ecological Ecotone” [4]. Since then, research on ecological vulnerability has gradually become a hot topic. In 2022, The Intergovernmental Panel on Climate Change (IPCC) pointed out that about 3.3 billion to 3.6 billion people worldwide live in an extremely vulnerable ecosystem to climate change [5]. The International Biological Program (IBP), the Man and the Biosphere Program (MAB), and the Geosphere Biosphere Program (IGBP) all consider ecological vulnerability as an important research area [6], and ecological vulnerability has become an important issue that the international community cannot avoid when facing ecological and environmental problems.

Since the reform and opening-up, China’s ecological environment has been increasingly affected by rapid industrialization and population urbanization, and ecological vulnerability has been increasing, and the overall quality and stability of the ecosystem are in an unpromising condition [7]. Based on the current situation of poor ecological fragility in China, in order to promote the development of ecological civilization construction, in recent years, the Chinese government has continuously improved the future territorial spatial development plan. On 23 May 2019, the CPC Central Committee and the State Council officially published “Several Opinions on Establishing the Supervision and Implementation of the Territorial Spatial Planning System”. Five levels and three categories of top-level designs of the land and space planning system were basically formed, of which five levels, respectively, correspond to China’s administrative management system, namely, national level, provincial level, municipal level, county level and township level. The national level focuses on strategy. While the township level focuses on implementation, while municipal land and space planning plays a key role in connecting the preceding with the following, which has both strategic and practical significance. Territorial Spatial Planning refers to the division of “three districts and three lines” and main functional areas by natural resources departments and related units. The purpose is to complete the planning and control of limited land resources by the state and to optimize the allocation of land resources scientifically and reasonably. At the same time, land space planning and environmental protection are closely related, directly affecting the country’s and region’s sustainable development. Rapid urbanization development means that human beings have acquired a lot of resources from nature. While using these resources to promote economic development also inevitably causes specific damage to the ecological environment. Therefore, sustainable development is the direction of our future development and a principle that should be followed by land space planning. At the same time, adhering to the idea of sustainable development in Territorial Spatial Planning guarantees the quality and efficiency of Territorial Spatial Planning. The evaluation results of ecological vulnerability of large and medium-sized cities can reflect the ecological status of urban areas with different functions as a complex ecosystem, and provide a basis for the delineation of three districts and three lines in the municipal Territorial Spatial Planning. The evaluation results can also reflect the relative vulnerability of ecological vulnerable areas, which is of great significance to the future ecological civilization construction of cities. The latest data show that China’s ecologically vulnerable areas of moderate or above account for 55% of the country’s terrestrial land space, of which 9.7% are extremely vulnerability areas and 19.8% are heavy vulnerability areas [8]. On this basis, how to accurately grasp the distribution status of ecologically fragile areas in large and medium-sized cities, combine various problems in the process of sustainable urbanization, scientifically and reasonably carry out targeted environmental treatment, and grasp the balance between economic and social development and ecological civilization construction is the key link for China to achieve high-quality economic development in the future.

Whether in the theoretical scope of ecological research or the applied research related to regional ecological security and ecological civilization construction, ecological vulnerability assessment has become an important research area of common concern in the world today [9]. The concept of ecological vulnerability was first proposed in Clements’ ecological staggered zone and further confirmed at the 7th SCOPE (Scientific Committee on Problems of the Environment) meeting in 1989 [10]. The existing definition of ecological vulnerability contains two aspects: the intrinsic vulnerability caused by the ecosystem within the ecosystem itself and the extrinsic vulnerability due to anthropogenic disturbances [11]. Ecological vulnerability, as an intrinsic property of ecosystems, only becomes apparent in response to external disturbances [12]. Ecological vulnerability assessment can effectively understand environmental changes, through the evaluation and analysis of ecological vulnerability, we can grasp the threat source of regional ecological vulnerability, understand the mechanism through which different influencing factors cause ecological vulnerability, and take corresponding measures according to the evaluation results and thus guide the rational use and effective protection of the regional ecological environment [13]. At the same time, there is a close relationship between ecological vulnerability and sustainable development. The fragility of the ecosystem often leads to the waste of resources, which will offset the advantages of sustainable economic growth and social development. At the same time, the realization of sustainable development also helps to reduce ecological vulnerability. Sustainable resource utilization and environmental protection measures can reduce the impact of activities on the ecosystem, thus reducing the ecological fragility caused by rapid urbanization.

Under the current background of ecological civilization construction, Territorial Spatial Planning is gradually becoming an important tool for practicing ecological civilization construction. Some scholars have studied the construction of the four-sector Territorial Spatial Planning governance system of “nature-government-market-society” under the background of ecological civilization era [14], some scholars have summarized and analyzed the construction of urban ecological network system in Territorial Spatial Planning [15], and some scholars have studied the compilation of special Territorial Spatial Planning in ecological civilization practice areas [16]. Most of the above-mentioned scholars’ research focuses on the macro level, that is, the academic research on the construction mechanism of the planning system or the overall planning. However, at present, there are few studies on the evaluation of urban ecosystem from a microscopic perspective, such as the ecological vulnerability evaluation of large and medium-sized cities. The results of such studies can provide some data support for the Territorial Spatial Planning of large and medium-sized cities and the delineation of “three districts and three lines”. The core of regional ecological vulnerability evaluation is the construction of the index system and the determination of the evaluation method. The selection of appropriate indicators through logical framework models is the most commonly used research tool, and the main logical framework models currently used are the “Pressure-State-Response” (PSR) model [17,18,19], the “Exposure-Sensitivity-Adaptive “ (ESA) model [20], and the “Sensitivity-Resilience-Pressure” (SRP) model [21], etc. Among them, the “sensitivity-resilience-pressure” model is based on the connotation of ecosystem stability, and its model structure is relatively comprehensive and systematically includes the basic elements of ecological vulnerability, which is widely used at present [22,23,24]. Regarding the evaluation methods of regional ecological vulnerability, fuzzy comprehensive evaluation method [25], comprehensive evaluation method [26], hierarchical analysis method [27,28], entropy weight method [29], grey clustering method [30], principal component analysis [20,31], etc. have been used. The research scope is mostly focused on ecological areas, such as watersheds [32,33,34,35,36,37], cities [2,38,39,40,41], mining areas [1,42], mountainous areas [43,44,45,46], islands [47,48], wetlands [49,50], protected areas, reservoirs [51] etc. Qi Shanshan et al. used a hierarchical analysis to explore the ecological vulnerability of watersheds based on the “ecological sensitivity-ecological resilience-ecological pressure” model [22]. Qi Yue used wetlands as the research object based on the “disturbance-sensitivity-restoration” model. Qi Yue took wetlands as the research object, constructed the ecological vulnerability evaluation index system based on the “disturbance-sensitivity-restoration” assessment framework, and explored the spatial distribution characteristics of ecological vulnerability of wetlands by using the interpolation method and comprehensive index method [20]. Zhang Huilin took mountainous areas as the research object, based on the “sensitivity-resilience-stress” model. Based on the “sensitivity-restoration-pressure” model, Zhang Huilin constructed an ecological vulnerability evaluation index system and applied spatial principal component analysis to evaluate the ecological vulnerability of mountainous areas [24].

Currently, China is in the stage of rapid urbanization, so the number of large and medium-sized cities is increasing, and the population scale is expanding. With the expansion of large and medium-sized cities, the ecological environment pressure faced by cities is rising. In this context, the environmental governance problem is especially prominent and fundamental. The significance of ecological vulnerability assessment for cities lies in that it can accurately identify the relatively fragile areas of urban ecology and the critical areas of environmental governance through the assessment results. However, the existing ecological vulnerability research mainly focuses on assessing natural regions, such as river basins. It is rare to evaluate the ecological vulnerability of cities as the research areas. A comprehensive and accurate assessment of the ecological vulnerability of large and medium-sized cities is particularly essential for promoting the sustainable development of cities and environmental protection. Shenyang, the only mega-city in the northeast of China, is a heavy industry city with more than 8 million population. Its highly concentrated population and economic conditions have triggered many problems with the city’s rapid development. Shenyang is China’s former heavy industry center and has developed industry without considering environmental pollution for several decades. At the same time, we have been conscious of a battery of problems arising. This development pattern has caused different urban development and transformation environmental governance issues. The rapid expansion of construction land has also generated many uncertain factors in Shenyang’s ecological environment, which has affected Shenyang’s ecological security. For example, we have recorded many floods in Shenyang in recent 20 years, and a grievous smog pollution incident occurred in 2015. It is an indispensable part of urban planning and sustainable development strategy to evaluate Shenyang’s ecological vulnerability and accurately realize the current situation of urban ecological vulnerability. Meanwhile, Shenyang’s ecological vulnerability assessment and environmental governance can also provide some reference for the sustainable development of cities with the same situation in China and even the world. Furthermore, the evaluation results of Shenyang’s ecological vulnerability are also effective in the layout of ecological civilization construction and Territorial Spatial Planning at the municipal level in Shenyang. As the central city of Northeast China, Shenyang’s Territorial Spatial Planning plays a connecting role. It will directly affect the sustainable utilization of resources and economic development model in Liaoning Province and other China’s northeastern provinces in the future as well. Combine the above two aspects, this paper selects Shenyang, a crucial central city in northeast China, as the research area and constructs an ecological vulnerability evaluation index system for Shenyang City based on the “Sensitivity-Resilience-Pressure” model, selecting 13 indicators from two dimensions of regional natural endowment and socio-economic development. Additionally, using spatial autocorrelation analysis and geographic probe model, the spatial distribution characteristics and driving factors of ecological vulnerability are further discussed to provide some support for the development of ecological civilization construction under the background of Territorial Spatial Planning, the delineation of “three districts and three lines”, especially the ecological red line. It gives policy references for large and medium-sized cities to optimize regional ecological patterns and achieve high-quality and sustainable economic and social development.

## 2. Materials and Methods

### 2.1. Study Area

Shenyang (41.20°~43.04° N, 122.42°~123.81° E) is located in the central part of Liaoning Province, which is an important connection between the Bohai Sea Rim and the northeast region (Figure 1). Shenyang has 13 county (district) level administrative regions, including 10 municipal districts, one county-level city and two counties. There are 9.073 million people lived in Shenyang at the end of 2020, with an urbanization rate of 84.52% and GDP of 657.16 billion yuan, accounting for 26.17% of the total in Liaoning Province. The northeastern part of Shenyang is a hilly and mountainous area, and the overall terrain is gradually flattened from the northeast to the southwest, and the landscape is gradually transitioned from hilly and mountainous to alluvial plains. Shenyang city belongs to the temperate semi-humid continental climate, with a large temperature difference throughout the year, and precipitation is mostly concentrated in summer, with 600–800 mm precipitation throughout the year. Shenyang city area has a wide variety of vegetation, and belongs to the intersection of Changbai-Mongolia-North China plant area.

### 2.2. Analytical Framework

Based on the “sensitivity-resilience-stress” model, this paper selects 13 indicators that can measure ecological vulnerability from different angles from three dimensions of ecological sensitivity, ecological resilience and ecological stress, as well as seven directions of topography, land surface, climate, vegetation, ecological vitality, economic development and human activities, and constructs an ecological vulnerability evaluation system. Through principal component analysis of indicators, the weights of each indicator are calculated, and the ecological vulnerability index map is drawn. The spatial and temporal distribution pattern is analyzed by visualizing the change trend, the spatial heterogeneity of ecological vulnerability in the study area is explored by using global/local autocorrelation tools, and the driving factors of the index system are analyzed by using geographic detectors. Finally, the ecological vulnerability of the study area in the future is predicted by CA-Markov model based on the existing data results. The research framework of this paper is shown in Figure 2.

### 2.3. Data Sources

Considering the availability of data and the process of urban development, the impact of ecological vulnerability in years that are too long away from the present is not particularly prominent. To comprehensively reflect the changing law of ecological vulnerability in Shenyang in recent ten years and the impact of urban development on the ecological environment, this study selected 2010–2020 as the research period and 2010, 2015 and 2020 as the time nodes, collected and analyzed the data, and evaluated the near future of Shenyang.

Topographic factors include elevation and slope. The digital elevation model (DEM) data of the study area is a 90 m × 90 m raster data of the geospatial data cloud. The slope data were obtained by analyzing the raster data using GIS 10.2 software.

Surface factors include landscape fragmentation and soil erosion: landscape fragmentation results were calculated using Fragstats 4.2 software with reference to land use types in the study area. Soil erosion intensity was selected from the modified generic soil loss equation (RULSE) model, with the expression:(1)$$S=\sqrt[{4}]{K\times \mathrm{LS}\times R\times C}$$
where, L is soil erosion sensitivity factor; K is soil erodibility factor; LS is slope length factor; R is rainfall erosion factor; C is vegetation cover factor.

Climatic factors include mean annual temperature and annual precipitation. Climate information was obtained from the China Meteorological Data Network study area and surrounding meteorological stations, and the meteorological data from the China Meteorological Administration were interpolated into 90 m × 90 m grid cell data using the Kriging method.

Vegetation factors include NDVI. normalized difference vegetation index (NDVI) was extracted from remote sensing images of Shenyang City using ENVI 5.4 software, which can accurately reflect the surface vegetation cover.

Ecological vitality factors including biological abundance were calculated on the basis of land use type data with reference to the study by Li Y. H. et al. [52].

The economic development factors are gross domestic product per capita and economic density. Data are obtained from Liaoning Statistical Yearbook and China County Statistical Yearbook.

Human activity factors include population density, nighttime light intensity, and land use type. Population density data were obtained from the LandScan dataset website with a resolution of 1000 m × 1000 m. Nighttime light intensity was obtained from the Harvard Dataverse platform. Land use type data were obtained from the Earth Big Data Science Project website.

### 2.4. Methods

#### 2.4.1. Evaluation Index System and Standardization

The Sensitivity-Resilience-Pressure (SRP) model adopted in this study is a comprehensive evaluation index system that can systematically evaluate a region. The SRP conceptual model is built with the stability of the ecosystem as the core. Among them, ecological sensitivity reflects the response to external interference caused by the relatively unstable factors existing in the internal environment of the system in the region, and this response, because of its lack of corresponding ability, leads to a certain degree of change in the development direction of the system, showing resilience. The degree of pressure comes from the external pressure of the system, mainly caused by human beings’ destruction of the environment. Ecological vulnerability is a vulnerable state caused by external pressure and insufficient ecosystem adaptability. Based on the analysis framework of ecological vulnerability evaluation system, the ecological vulnerability evaluation index system was constructed with reference to relevant studies (Table 1). Due to the differences in the order of magnitude of index data, the research data were processed by the method of extreme difference standardization before the ecological vulnerability analysis and evaluation. The standardization formula is as follows:(1)Positive indicators:
(2)$$A_{i}=\frac{a_{i}-a_{\min}}{a_{\max}-a_{\min}}$$

(2)Negative indicators:

(3)$$A_{i}=\frac{a_{\max}-a_{i}}{a_{\max}-a_{\min}}$$
where, *i* is the number of indicators, $A_{i}$ is the standardized value of the *i-*th indicator, which ranges from 0 to 1; a is the data of the *i*-th indicator; $a_{\min}$ is the minimum value of the *i*-th indicator, $a_{\max}$ is the maximum value of the *i*-th indicator. Ecological vulnerability becomes higher with increasing values of positive indicators or decreasing values of negative indicators. Among them, average annual temperature, annual precipitation, NDVI and biological abundance are negative indicators, and the rest are positive indicators except land use type. The land use type is the type variable, and the vulnerability of different land types is assigned to them according to the treatment of related studies, and the assignment results are shown in Table 2.

#### 2.4.2. Principal Component Analysis

Principal component analysis can reflect information with fewer integrated indicators and maximize the retention of information reflected by more original variables. In this paper, the original indicators are replaced by the principal components whose cumulative contribution rates meet the requirements, and then the principal component indicators are used to determine the principal factors involved in the evaluation, and finally the composite indicators of the principal components are calculated on the basis of:(4)$$F_{i}=a_{1i}X_{1}+a_{2i}X_{2}+\ldots +a_{pi}X_{p}$$
where, $F_{i}$ is the *i*-th principal component; $a_{1i}$, $a_{2i}$, *…*$a_{pi}$, is the feature vector of each indicator corresponding to the i-th principal component; $X_{1}$, $X_{2}$, *…*,$X_{p}$ is the factor of each indicator.

The ecological vulnerability index (EVI) is calculated from the results of the principal components, and its basic formula is:(5)$$I_{\mathrm{EV}}=\alpha_{1}F_{1}+\alpha_{2}F_{2}+\cdots +\alpha_{N}F_{N}$$
where, $I_{\mathrm{EV}}$ is the ecological vulnerability index; $F_{1}$, $F_{2}$, *…*, $F_{N}$ is the 1st, 2nd,…, N-th principal component; $\alpha_{1}$, $\alpha_{2}$, …, $\alpha_{N}$ is the contribution rate corresponding to the 1st, 2nd,…, N-th principal component; N is the number of retained principal components. A higher value indicates a more vulnerable ecosystem.

#### 2.4.3. Spatial Autocorrelation

In this paper, we use the spatial autocorrelation analysis module in GeoDa software to explore the spatial correlation of the study area. Among the obtained analysis results, positive correlation indicates that the attributes of the adjacent spatial units at the boundaries within the study area have the same evolutionary trend and there is a certain spatial clustering between them. On the contrary, it indicates that the attributes of adjacent spatial units at the boundary within the study area are different and there is no spatial clustering.

(1)Global spatial autocorrelation

Global spatial autocorrelation is mainly used to describe the spatial distribution and clustering characteristics of a certain attribute within the whole study area. At present, Moran′s I index is widely used in spatial autocorrelation studies, and the calculation formula is:(6)$$\mathrm{Moran’s}\;I=\frac{n\sum_{i=1}^{n}\sum_{j=1}^{m}W_{ij}\left(x_{i}-\bar{x}\right)\left(x_{j}-\bar{x}\right)}{\sum_{i=1}^{n}\sum_{j=1}^{m}W_{ij}\sum_{i=1}^{n}{\left(x_{i}-\bar{x}\right)}^{2}}$$
where, $x_{i}$ is the observation of the *i*-th raster; n is the number of rasters; $W_{ij}$ is the binary adjacency space weight matrix; *i* = 1, 2,…, *n*; j = 1, 2,…, *m*. The Moran′s I index ranges from (−1,1), and when the index value is greater than 0, it means that the attribute exhibits a spatially aggregated state; less than 0, it means that the attribute is in a discrete state in spatial distribution; tending to 0, it means that the attribute is in a randomly distributed state in space.

(2)local spatial autocorrelation

Local spatial association index (LISA) is mainly used to reflect the degree of correlation between the indexes in the local area and the neighboring areas. The local Moran′s I index is more commonly used to analyze the local spatial variation characteristics of the study area and is calculated as follows:(7)$$\mathrm{Moran}’\;\mathrm{sI}=\frac{\left(x_{i}-\bar{x}\right)\sum_{j=1}^{m}w_{ij}\left(x_{j}-\bar{x}\right)}{\frac{1}{n}\sum_{i=1}^{n}{\left(x_{i}-\bar{x}\right)}^{2}}$$
where, *n* is the number of analysis units; $w_{ij}$ is the elements of the spatial weight matrix; $x_{i}$, $x_{j}$ is the spatial units after row normalization.

The results of the local spatial autocorrelation analysis were classified into five categories using Geoda software: “HH” high-high clustering, “HL” high-low clustering, “LH “LH” low-high clustering, “LL” low-low clustering, and “NN” non-significant type.

#### 2.4.4. GeoDetector

GeoDetector is a research model that detects the spatial heterogeneity of regional geographic factors and explains the driving forces affecting the distribution of such factors. The model consists of four modules: factor detector is used to examine the variability of the dependent variable and the degree of variability of the independent variable within the study space, and the detection result is measured by the q value, with a higher q value indicating that the distribution of the dependent variable within the study space is deeply influenced by the independent variable. Interaction detector is used to identify whether the interaction between different factors has enhanced the explanation of the dependent variable. Risk detector is used to identify the attributes between any two sub-regions within the study area. The risk detector is used to identify whether there is a significant difference in the mean values of attributes between any two sub-regions within the study area, as measured by the t-statistic. The ecological detector is used to compare whether there is a significant difference in the influence of two independent variables on the spatial distribution of the dependent variable, as measured by the F-statistic.

In this paper, the factor detector module and the interaction detector module are selected to investigate the relationship between the selected factors and the spatial distribution pattern of ecological vulnerability in Shenyang, in which the factor detector is used to detect the degree of influence of the selected factors on the spatial distribution pattern of ecological vulnerability. The interaction detector is used to identify the interaction between the influencing factors and their explanation of the spatial distribution pattern of ecological vulnerability in the area.

#### 2.4.5. CA-Markov Model

In this study, the CA-Markov model is used for the simulation, prediction and analysis of ecological vulnerability. The Markov model was first proposed by Andrey Markov, a mathematician of the former Soviet Union. It is a process theoretical model based on the Markov random process system so as to achieve the purpose of prediction and random control. The cellular automata (CA) model is a lattice dynamics model with discrete temporal and spatial states. It focuses on the interaction of cells with different temporal and spatial characteristics. It has strong spatial computing and simulation ability and is especially suitable for the dynamic simulation and spatial display of self-organizing functional systems. The formula is expressed as follows:(8)$$S_{t\;+1}=f\left(S_{t},N\right)$$
where S is a set of cellular states; N is the cellular field; t and t+1 are different moments; f is the cellular transformation rule of local space.

The Markov model is a stochastic model in the time domain, which is transformed to the state at t + 1 time according to the state of the event at t time, and the state at t + 1 time is only related to the state at t time. Its essence is to predict the probability of an event. The transition matrix is a digital reflection of the possibility of events transforming into the t + 1 state at t time, and it is an important quantitative basis for simulation and prediction results under the Markov model. The formula is expressed as follows:(9)$$S_{t+1}=P_{\mathrm{ij}}\times S_{t}$$
where $P_{\mathrm{ij}}$ is a state transition matrix, and its formula is expressed as follows [53]:(10)$$P_{\mathrm{ij}}=\left[\begin{array}{cccc} P_{11} & P_{12} & \cdots & P_{1n}\\ P_{21} & P_{22} & \cdots & P_{2n}\\ \cdots & \cdots & \cdots & \cdots \\ P_{n1} & P_{n2} & \cdots & P_{nn} \end{array}\right]$$
where 0 ≤ $P_{\mathrm{ij}}$ < 1, (i,j = 1,2, …, n); n is the number of land-use types; $P_{\mathrm{ij}}$ represents the probability that the initial type i is converted to type j; i is the row of the matrix; j is the column of the matrix. Each row of the matrix represents the probability of land use type i transforming to each land-use type.

The CA-Markov model combines the ability of CA models to simulate spatial changes with the ability of Markov models to predict long-term time series, which provides good stability in simulating land use change models and also avoids the occurrence of stochastic distributions in the simulation process, enhancing the accuracy of the simulation. This study uses the CA-Markov model in IDRISI software to simulate and predict the ecological vulnerability of the study area in 2020 based on the ecological vulnerability data of the study area in 2010 and 2015, and then uses the CROSSTAB tool in the software to overlay the actual data and the simulated data in 2020 to analyze the accuracy of the simulation results and verify the ecological vulnerability of the study area The CA-Markov model was used to analyze the accuracy of the simulation results and verify the applicability of the ecological vulnerable model in the study area.

## 3. Results

### 3.1. Spatial Distribution Characteristics of Ecological Vulnerability

Principal component analysis was used to analyze 13 evaluation factors in the ecological vulnerability evaluation system of Shenyang City, and the first 6 principal components with a cumulative contribution rate of more than 85% were screened out, and the results are shown in Table 3. The ecological vulnerability index of the study area was calculated according to Equation (5), and combined with the natural breakpoint method, the ecological vulnerability grading criteria of Shenyang city were classified (Table 4), and the ecological vulnerability was classified as slight vulnerability (0.091~0.226), light vulnerability (0.226~0.262), medium vulnerability (0.262~0.303), heavy vulnerability (0.303~0.368) and extreme vulnerability (0.368~0.564) 5 levels.

According to the calculated values of ecological vulnerability index in three years, the average values of ecological vulnerability index in Shenyang in 2010, 2015 and 2020 are all about 0.25, and the values are not much different. During these ten years, the ecological vulnerability of Shenyang is generally in a relatively stable state, with the lowest ecological vulnerability index in 2010 being 0.255 and the highest ecological vulnerability index in 2015 being 0.262. The distribution of ecological vulnerability in the three years is basically the same, and the areas of light and medium ecological vulnerability are relatively large. The areas of light and medium vulnerability each year account for more than 60% of Shenyang’s area, among which the areas of light and medium vulnerability reached 75% in 2015. The proportion of extremely vulnerable areas is at a low level in the selected study years. In 2020, the proportion of extremely vulnerable areas is only 2.73%, while in 2010, the proportion of extremely vulnerable areas was only 1.32%, which shows that there are very few ecologically vulnerable areas in Shenyang. The corresponding area of the least vulnerable slight vulnerable area was 20% in three years, and only the slightly vulnerable area in 2015 is relatively small, only 13.16%.

From the spatial distribution pattern of ecological vulnerability (Figure 3), the spatial distribution of ecological vulnerability in Shenyang has been relatively stable from 2010 to 2020, and the areas with high ecological vulnerability have been concentrated in the main urban area during this decade. The ecological vulnerability index of Shenhe District is the highest in the selected three years, while Tiexi District, Dadong District, Huanggu District and Heping District also have large areas with relatively vulnerable ecology. These areas have low vegetation coverage and high degree of landscape fragmentation. The areas with the best ecological vulnerability have been concentrated in Liaozhong District in the southwest of Shenyang and Sujiatun District in the south. In the past ten years, Liaozhong District and Sujiatun District have also maintained good ecological conditions, while Faku and Kangping in the north of Shenyang have some medium vulnerability areas and slight vulnerability areas. Generally speaking, the ecological vulnerability of Shenyang presents a spatial distribution pattern in which the suburbs are superior to the urban areas, and the southern areas are superior to the northern areas, which is mainly caused by the strong interference of human activities on the ecosystem of urban built-up areas. From the time scale, from 2010 to 2020, the area of ecologically extreme vulnerability areas in the main urban area has obviously increased and expanded. With the expansion of the city and economic development, the ecosystem around the city inevitably develops in a vulnerable direction, and some areas with good ecological conditions around the original urban area have also begun to turn into moderately fragile areas. However, the ecological vulnerability of Kangping and Faku areas in the north of Shenyang has obviously improved. In 2010, some of the medium vulnerability areas and heavy vulnerability areas in the northwest of Kangping changed into light vulnerability areas and medium vulnerability areas in this decade. The areas with better ecological vulnerability in Liaozhong and Sujiatun in southern Shenyang have also been maintained at a good level. Compared with other areas in Shenyang, these areas have stronger self-recovery ability and anti-interference abilities, and the regional ecosystem is more stable. The total share of ecological vulnerability in each district and county of Shenyang in each year is shown Table 5.

### 3.2. Spatial Correlation Analysis of Ecological Vulnerability

The analysis used a 900 m × 900 m scale raster to analyze the characteristics of global autocorrelation and local clustering of ecological vulnerability index in Shenyang city with the help of Geoda software. From the results of global Moran’s I scatter distribution (Figure 4), it can be seen that the global Moran’s I index of ecological vulnerability in Shenyang City in 2010, 2015 and 2020 are 0.867, 0.813 and 0.841, respectively, and the *p*-values are less than or equal to 0.05. The significance is high, which indicates that ecological vulnerability in Shenyang City does not occur randomly, and there is a positive correlation in space with obvious positive clustering characteristics.

The spatial clustering characteristics of ecological vulnerability in Shenyang were analyzed by local autocorrelation, and the LISA spatial distribution map of ecological vulnerability index in Shenyang was obtained in three periods (Figure 5). The HH agglomeration area accounts for 25.15%, which is concentrated in the main urban area and occurs in a row, while HH agglomeration areas also appear in Kangping County, northern Faku County, eastern Yuhong District, western Hunnan District and southern Shenbei New District, while LH agglomeration and HL agglomeration areas appear in fewer areas and account for no more than 1% of the total area, and there is little difference between the two periods in 2015 and 2020, with low and low agglomeration areas distributed in Liaozhong District, Sujiatun District and southwestern Xinmin City. The area share decreased significantly to 21.36% and 22.34%, respectively. The high-high agglomeration area decreases compared with 2010, with 12.30% and 12.76%, respectively, mainly distributed in the main city center, eastern Yuhong District, western Hunnan District and southern Shenbei New District, while Kangping County and northern Faku County transform into insignificant areas; LH agglomeration and HL agglomeration area remain less sporadic and point-like distribution. The non-significant areas increased more, by 18.07% and 17.98%, respectively. The data from the three periods show that in Liaozhong District, Xinmin City and Sujiatun District, patches with low ecological vulnerability are spatially clustered, resulting in maintaining a low level of regional ecological vulnerability. On the contrary, the main city center, Yuhong District, Hunnan District and Shenbei New District are mainly influenced by patches with high state vulnerability, while Kangping County and Faku County are transformed from high HL clustering areas to non-significant areas, with no significant regional clustering.

### 3.3. Analysis of Factors Influencing Ecological Vulnerability

In order to investigate the main influencing factors of ecological vulnerability in Shenyang city, a geographic detector model was used to analyze the independent variable as 10 ecological vulnerability evaluation factors and the dependent variable as ecological vulnerability index.

The results of factor detector detection are shown in Table 6, and the *p*-values of all factors are close to 0, while the differences of q-values are obvious, indicating that there is no prominent single factor to explain the ecological vulnerability in Shenyang city, and there are more influential factors. 2010 index factors on the spatial variability of ecological vulnerability of Shenyang city are ranked from the largest to the smallest in terms of explanatory power as average annual temperature, NDVI, night light intensity, elevation, economic density, biological abundance, annual precipitation, population density, GDP per capita, land use type, landscape fragmentation, slope and soil erosion, with q-values of 0.356, 0.326, 0.295, 0.292, 0.261, 0.232, 0.221, 0.200, 0.182, 0.171, 0.096, 0.077 and 0.050, respectively. The explanatory power of ecological vulnerability impact factors in 2015, from largest to smallest, were night light intensity, NDVI, economic density, biological abundance, land use type, GDP per capita, population density, landscape fragmentation, average annual temperature, elevation, annual precipitation, soil erosion, and slope, and their q-values were 0.506, 0.477, 0.458, 0.415, 0.326, 0.285, 0.257, 0.164, 0.140, 0.102, 0.099, 0.034, and 0.032. The explanatory power of ecological vulnerability impact factors in 2020, from largest to smallest, are nighttime light intensity, economic density, NDVI, biological abundance, GDP per capita, land use type, population density, average annual temperature, annual precipitation, landscape fragmentation, elevation, The top five factors with higher q-values in all three periods were selected to show that NDVI, nighttime light intensity, biological abundance, economic density, and economic density are the most important factors in the long term. Population density, biological abundance and economic density are the main influencing factors affecting the change of ecological vulnerability in Shenyang. The other index factors explained the ecological vulnerability of Shenyang to a lesser extent.

The joint effect of multiple factors on ecological vulnerability may differ from the effect of a single factor. The interaction detector results contain five kinds of results: non-linear weakening, single-factor non-linear weakening, independent, non-linear enhancement and two-factor enhancement, and the interaction detector model is used for ecological vulnerability evaluation in Shenyang City to determine whether the joint effect of two factors increases or weakens the explanatory power of ecological vulnerability compared with the single-factor effect. The results of interaction probe detection are shown in Figure 6, Figure 7 and Figure 8. The 13 evaluation factors of ecological vulnerability in Shenyang in 2010, 2015 and 2020 produced 78 interaction synergy results after interaction detection, mainly 2 types of two-factor enhancement and non-linear enhancement, which indicated that the interaction between any two factors had stronger effects on regional ecological vulnerability than the single factor effect.

The strongest interaction between annual mean temperature and NDVI bivariate in 2010 with q(X1∩X2) value of 0.739, followed by the interactions between annual mean temperature and nighttime light intensity bivariate, annual mean temperature and biological abundance bivariate, annual mean temperature and economic density bivariate, elevation and NDVI bivariate, and The interaction between annual average temperature and land use type, q(X1∩X2) values are 0.694, 0.671, 0.669, 0.632 and 0.616, respectively, that is, the interaction between NDVI, annual average temperature, nighttime light intensity, biological abundance, economic density, elevation and land use type has a strong impact on ecological vulnerability in Shenyang in 2010.

The strongest interaction between the two variables of annual average temperature and NDVI was found in 2015 with q(X1∩X2) value of 0.669, followed by the interaction between the two variables of annual average temperature and nighttime light intensity, the interaction between the two variables of landscape fragmentation and nighttime light intensity, the interaction between the two variables of NDVI and elevation, the interaction between the two variables of NDVI and GDP per capita with q(X1∩X2) values of 0.610, 0.603, 0.602, 0.599 and 0.590, respectively. Interaction, NDVI and economic density interaction, q(X1∩X2) values are 0.610, 0.603, 0.602, 0.599 and 0.590, respectively, that is, the interactions of NDVI, average annual temperature, nighttime light intensity, landscape fragmentation, elevation, GDP per capita and economic density have strong effects on ecological vulnerability in Shenyang in 2015.

The strongest interaction between annual mean temperature and NDVI bivariate was found in 2020 with q(X1∩X2) value of 0.712, followed by the interaction between annual mean temperature and nighttime light intensity bivariate, NDVI and GDP per capita bivariate, NDVI and elevation bivariate, annual mean temperature and economic density bivariate, and The interaction between annual average temperature and biological abundance, q(X1∩X2) values are 0.647, 0.640, 0.636, 0.623, 0.605, respectively, that is, the interaction between NDVI, annual average temperature, nighttime light intensity, GDP per capita, elevation, economic density, and biological abundance in 2020 has a strong impact on ecological vulnerability in Shenyang.

The results of the integrated factor detector show that NDVI, economic density and nighttime light intensity are the key driving factors of ecological vulnerability in Shenyang. Therefore, when utilizing and developing ecological resources, promoting socio-economic development and urbanization in Shenyang, the influence of regional natural resource endowment, socio-economic conditions and the combined effect of these factors on ecological environment should be considered to strengthen large efforts to protect ecologically vulnerable areas and pay attention to the existing ecological problems in time to promote the harmonious development of human and nature.

### 3.4. Simulation Analysis of Ecological Vulnerability in Shenyang in 2025

With the aid of CA-Markov model in IDRISI software, the spatial layout of ecological vulnerability in Shenyang in 2020 was simulated based on the ecological vulnerability area transfer matrix of 2010–2015 and the ecological vulnerability suitability image set of Shenyang City in 2015 as the base period image, and the simulation results are shown in Figure 9.

In this paper, Kappa coefficient is used to verify whether the simulated data of Shenyang in 2020 is consistent with the current situation data and to quantitatively evaluate the accuracy of model simulation. If Kappa ≥ 0.75, then it indicates that the reliability of the simulation results is high; if 0.4 ≤ Kappa ≤ 0.75, then it indicates that the simulation is average; if 0.4 ≤ Kappa, then it indicates that the simulation is poor, and the calculation formula is shown as follows.
(11)$$\mathrm{Kappa}=\frac{p_{0}-p_{c}}{p_{p}-p_{c}}$$
where *p*_0_ is the ratio of the number of correctly simulated grids to the total number of grids, *p_p_* is the ratio of the number of correctly simulated grids to the total number of grids under ideal conditions, and *p_c_* is the ratio of the number of correctly simulated grids to the total number of grids under random conditions.

The spatial superposition of Shenyang’s 2020 status quo data and the 2020 simulation results yielded a Kappa coefficient of 0.721, which is close to 0.75, then it means that the consistency between the simulated data map of ecological vulnerability in Shenyang in 2020 and the status quo data map is relatively high and the simulation is better, i.e., the CA-Markov model is more effective in simulating the ecological vulnerability of Shenyang The feasibility of CA-Markov model in simulating the spatial layout of ecological vulnerability in Shenyang is better.

Based on the ecological vulnerability spatial pattern map of Shenyang in 2020, the ecological vulnerability transfer matrix from 2015 to 2020 is input, and the ecological vulnerability pattern of Shenyang in 2025 is predicted and simulated based on the ecological vulnerability suitability image set of Shenyang with 5 cycles, and the results are shown in Figure 9. From the ecological vulnerability results in 2025, it can be seen that the ecological vulnerability of Shenyang in 2025 is light vulnerability and medium vulnerability are dominant, and in terms of quantity, the percentages of area of slight vulnerability, light vulnerability, medium vulnerability, heavy vulnerability and extreme vulnerability are 28.38%, 43.58%, 19.74%, 5.42% and 2.88%, respectively.

The results were imported into ArcGIS software to compare the 2025 ecological vulnerability simulation results with the 2020 status quo data, and the area change and transfer results were obtained as shown in Figure 9 and Table 7: By comparing the 2025 ecological vulnerability simulation results with the 2020 status quo data in Shenyang, it can be seen that in the simulation results in 2025, the area of the slight and extreme vulnerability areas increased, among which the slight vulnerability area In the simulation results of 2025, we can see that the area of slight vulnerability area and extreme vulnerability area will increase, among which slight vulnerability area will increase by 1029.040 km^2^, or 8.00%, and extreme vulnerability area will increase by 18.808 km^2^, or 0.15%. Meanwhile, the area of light, medium and heavy vulnerability area will decrease slightly, among which light vulnerability area will decrease by 193.175 km^2^, or 1.50%, medium vulnerability area will decrease by 620.233 km^2^, or 4.82%, and heavy vulnerability area will decrease by 234.440 km^2^, or 4.82%. Decrease 234.440 km^2^, accounting for 1.82% decrease

From the transfer matrix (Table 8), in 2025, compared with 2020, the transfer out of the slightly vulnerable area to the lightly vulnerable area is more, with an area of 14.495 km^2^, accounting for 88.27% of the transferred out area, while the main transfer in part all comes from the lightly vulnerable area, with an area of 1045.461 km^2^. The transfer out of the lightly vulnerable area to the lightly vulnerable area is more, with an area of 1045.461 km^2^, accounting for 95.13% of the transferred out area, while the main transfer in part comes from the lightly vulnerable area and the moderately vulnerable area, with an area of 1.60% and 98.40%, respectively. In this case, 95.13%, while the main part of the transfer comes from the slightly vulnerable area and the moderately vulnerable area, with an area of 14.495 km^2^ and 891.258 km^2^, respectively, accounting for 1.60% and 98.40% of the transferred area. The moderately vulnerable area is transferred to the lightly vulnerable area more often, with an area of 891.258 km^2^, accounting for 99.66% of the transferred area, while the main part of the transfer mainly comes from the heavily Vulnerable areas, with an area of 218.713 km^2^, accounting for 79.79% of the total area. More heavily vulnerable areas were transferred out to moderately vulnerable areas, with an area of 218.713 km^2^, accounting for 91.59% of the transferred area, while the main transferred-in parts all originated from moderately and extremely vulnerable areas and were less, with an area of 3.081 km^2^ and 1.266 km^2^, respectively. Less extremely vulnerable areas were transferred out to heavily. The area of 1.266 km^2^ was transferred from the extremely vulnerable area, while the area of 20.075 km^2^ was mainly transferred from the heavily vulnerable area, with little change.

With the gradual social and economic development of Shenyang, the demand for land in urban space and agricultural space is increasing, and their land area continues to increase, and the land surface hedge of ecological space will decrease. This irregular expansion of urban space and agricultural space will easily damage the regional ecosystem, which will cause a significant reduction in the land area of ecological space and undermine the stability and coordination of development. Shenyang, as an old central city, has superior geographical conditions and better ecological environment, creating an important ecological status of Shenyang. Therefore, in the context of Shenyang’s economic and social development, it is important to avoid the destruction of the environment and pay attention to ecological and environmental protection in order to achieve sustainable and coordinated development of society, economy and ecology.

## 4. Discussion

### 4.1. Distribution Pattern of Ecological Vulnerability in Shenyang

In this study, the related factors such as elevation, slope and soil erosion degree were selected to construct the evaluation index system of ecological vulnerability in Shenyang, and the ecological vulnerability of Shenyang was evaluated by spatial analysis and principal component analysis. From the spatial distribution pattern of ecological vulnerability, the areas with higher ecological vulnerability in Shenyang are concentrated in Huanggu District, Tiexi District, Heping District, Dadong District and Shenhe District, while the areas with lower ecological vulnerability are scattered around the urban areas, and some areas with higher ecological vulnerability are also found in Kangping and Faku in the north. The overall spatial distribution pattern of ecological vulnerability is relatively stable from 2010 to 2020, except that the ecological vulnerability of Kangping and Faku has a trend of improvement in ten years, and the ecological vulnerability level of other areas has not changed much. The main reason is that the urban area is the key area of human production and life. In this area, the vegetation coverage is low, the landscape fragmentation is high, the population density is high, and the land use type is relatively single, resulting in extremely vulnerable ecology. However, the areas with low ecological vulnerability are mostly suburban areas far away from urban areas, such as Liaozhong and Xinmin. Because of their high vegetation coverage, complex landscape structure, low landscape fragmentation and multiple ecological functions, their ecological environment is relatively vulnerable. Overall, the ecological vulnerability of Shenyang shows a trend of increasing from south to north and from west to east in space.

### 4.2. Simulation and Prediction of Ecological Vulnerability in Shenyang

Based on CA-Markov model, this study predicted the distribution pattern of ecological vulnerability in 2025 based on the evaluation results of ecological vulnerability in 2010, 2015 and 2020, and the results basically conformed to the spatial distribution pattern of ecological vulnerability in Shenyang in the previous decade. By comparing the prediction results in 2020 with the evaluation results in 2020, the Kappa coefficient was 0.72. Combined with the above two points, it can be judged that the prediction results can meet the requirements for the future. As a result, most of the extremely vulnerable areas in Shenyang are still distributed in the five districts of the city, while the ecological vulnerability outside the urban area remains at a low level. By comparing the area change and transfer results obtained from the predicted data with the current data, it can be seen that the area ratio of extremely vulnerable areas has increased slightly, and the increased areas are still located near the urban areas. Through analysis, the possible reason is that with the development of social economy, the demand for land in urban space continues to increase, resulting in the expansion of cities, which may lead to the transformation of the original ecological vulnerability areas around the urban areas into extreme vulnerability areas. This phenomenon also requires us to pay attention to the importance of sustainable development, take environmental control measures in the process of development, always implement the concept of sustainable development in the process of development, pay attention to the protection of ecological space while accelerating the urbanization process, and avoid further deterioration of the original ecologically fragile areas. In addition, the CA-Markov forecast in this study does not set a certain scenario, that is, it is assumed that the existing policy conditions will remain unchanged, but the overall forecast results are still reliable.

### 4.3. Policy Recommendations

Due to the difference of regional natural resource endowment and economic development level, the ecological vulnerability of Shenyang’s internal regions presents obvious spatial differentiation characteristics, and the phenomenon of contiguous ecological vulnerability regions becomes more and more prominent. Under the guidance of the dual goals of comprehensively guaranteeing the construction of regional ecological civilization and realizing the high-quality development of urban economy and society, we must attach importance to the assessment of regional ecological vulnerability, strengthen the zoning of regional ecological functions in combination with the land and space planning, and protect and develop regional ecological resources in a targeted manner. It is suggested that measures should be taken to prevent disasters and reduce the interference of human activities on the ecological environment in important ecologically fragile and ecologically sensitive areas. At the same time, in areas with severe soil erosion, we should do an excellent job in soil and water conservation and gradually and orderly implement returning farmland to forests. Under the guidance of the dual goals of comprehensively guaranteeing the construction of regional ecological civilization and realizing the high-quality development of urban economy and society, we must attach importance to the assessment of regional ecological vulnerability, strengthen regional ecological function zoning in combination with land space planning, and protect and develop regional ecological resources in a targeted manner. Although the central urban area plays a crucial role in economic development and social progress, considering that the ecological fragility of the central urban area is more fragile than that of the suburbs, it is suggested that a certain proportion of ecological functional areas, such as parks and green spaces should be reserved in the central urban area in the process of urban development planning. In the process of social and economic development, attention should be paid to protecting the ecological environment, and the uncontrolled expansion of construction land should not be allowed, which will lead to the destruction of the regional ecosystem and the substantial reduction in the originally small ecological space land area. In the process of urban development, we should consistently implement the ecological priority strategy, strengthen the construction of urban ecological civilization, and find a balance between ecological development and economic development. At the same time, in the process of sustainable urbanization, it is also necessary to scientifically consider how to coordinate the relationship between reducing the impact of urbanization on the ecosystem and promoting the needs of urban development according to the results of ecological vulnerability assessment. For example, the central area of a city plays a key role in economic development and social progress, but it is precisely because of the rapid economic development that the ecological vulnerability in the city is obviously stronger than that in the suburbs. Therefore, sustainable planning and construction methods, such as ecological architecture, water-saving technology and low-carbon urban planning, can be adopted to alleviate the impact of the central area on the urban ecosystem. In areas with good ecological fragility such as suburbs, the focus of development can be placed on economic construction, while environmental protection measures can be taken to ensure that the urbanization process will not cause a devastating blow to the local ecological environment, and at the same time, the uncontrolled expansion of construction land cannot be allowed. It is necessary to determine the ecological red line in the urbanization process, strengthen the construction of ecological civilization, and find the balance between ecological development and economic development to promote the sustainable development of the city.

### 4.4. Limitations and Future Prospects

Based on the conceptual model, this study selected the influencing factors to evaluate the ecological vulnerability of Shenyang in recent ten years, and analyzed and simulated the driving factors with the help of geographic detector and CA-Markov model. The results can provide some theoretical support for the land space planning of Shenyang and the demarcation of three districts and three lines. Of course, the following research needs to be strengthened in the future: (1) If the model used in this paper is combined with Delphi method, the model results will be more perfect. (2) Due to the limitation of spatial resolution and interpolation technology, the index has certain uncertainty. Precipitation, temperature and other indicators that interpolate the original data of weather stations have certain limitations in representing ecological vulnerability. Due to the lack of spatial resolution, the data of population density may also affect the accuracy of the results. Due to the limitation of data sources, some indicators are not included in the index system of this study, and the results of ecological vulnerability assessment will be inaccurate due to the selection of indicators. (3) The CA-Markov model used in this study was originally applied to the simulation of land use change, and the simulation of ecological vulnerability in this study is also a new attempt. The results show that the simulation accuracy is high, which shows that the application of this model in this paper is reasonable. However, the prediction in this study does not set a certain scenario, but only assumes the future development of ecological vulnerability pattern under the existing natural environment and policy conditions. Therefore, there is a certain error in the high probability of the prediction results. It is the direction that we need to work hard in the future to set different development scenarios and then predict the ecological vulnerability pattern under different scenarios to improve the prediction accuracy of regional ecological vulnerability.

## 5. Conclusions

(1) In 2010, 2015 and 2020, Shenyang’s ecological vulnerability was in a state of lightly vulnerability areas. From 2010 to 2020, the ecological vulnerability index generally showed a trend of first increasing and then decreasing. Generally speaking, the ecological vulnerability of Shenyang presents a spatial change trend from south to north and from west to east. Extreme vulnerability and heavy vulnerability are mainly concentrated in the central and eastern urban areas of Shenyang, while slight vulnerability and light vulnerability are mainly located in the southwest areas with good natural environment and less human disturbance.

(2) There are significant differences in ecological vulnerability indices among 13 districts and counties in Shenyang. Shenhe District, Heping District, Tiexi District and Dadong District have been maintained in extremely vulnerable state, while Huanggu District evolved from predominantly extreme vulnerability in 2010 to heavy vulnerability in 2015, and then converted back to extreme vulnerability in 2020, with the mean value of vulnerability index decreasing and increasing. The ecological vulnerability of these five districts has remained at a high level because these five districts are the central urban areas of Shenyang, which are greatly affected by indicators such as GDP and night light intensity, the ecological damage is severe, and the anti-interference ability is weak. Hunnan District converted from medium to heavy vulnerability in 2015, and then converted back to medium vulnerability in 2020. This phenomenon is mainly affected by vegetation coverage and night light index. The construction of Hunnan New District started to accelerate after 2010, and the environmental protection problem was neglected during the construction process. However, after 2015, this situation was improved due to the adoption of various ecological protection measures. Kangping County and Yuhong District has been in the same level of medium vulnerability. Faku County changed from medium vulnerability to light vulnerability status in 2020. Although the urbanization level in these areas is not high, and the impact of human activities on the environment is less, the ecological vulnerability level is maintained at a medium vulnerability level due to land use types and climate factors. Shenbei New District, on the other hand, converted from light to medium vulnerability in 2015, and the degree of vulnerability increased. Xinmin City has been in the same level of light vulnerability, Liaozhong District changed from slight vulnerability to light vulnerability, and then back to slight. Sujiatun District changed from slight vulnerability to light vulnerability. The ecological vulnerability index value increases. Xinmin city, Shenbei New District, Liaozhong District, and Sujiatun District have slow economic development, low urbanization level, high vegetation coverage and low population density. The impact of human activities on the ecological environment in these areas is smaller than that in the main urban areas, so the ecological vulnerability index is generally maintained at a low level and will fluctuate slightly.

(3) Spatially, in Liaozhong District, Xinmin City and Sujiatun District, patches with low ecological vulnerability are spatially clustered, resulting in maintaining a low level of regional ecological vulnerability. On the contrary, the main city center, Yuhong District, Hunnan District and Shenbei New District are mainly affected by patches with high ecological vulnerability. Meanwhile, Kangping County and Faku County are transformed from high-high clustering areas to non-significant areas, and there is a trend of ecological vulnerability turning good.

(4) The GeoDetector results showed that compared with the role of single factor, the ecological vulnerability of Shenyang city area was more susceptible to the interaction between factors, and NDVI, economic density and nighttime light intensity were the key drivers of ecological vulnerability in Shenyang city. In view of this, the ecological construction development about Shenyang should be considered comprehensively, and the improvement of a single environmental problem should not be the ultimate goal of ecological construction, and the construction of regional ecological environment optimization should be promoted in a coordinated manner while responding positively to the ecological vulnerability of key regions.

(5) The CA-Markov model predicts better results, and the ecological vulnerability of Shenyang City will increase in 2025 in the area of slightly vulnerability areas and extreme vulnerability areas, and the area of slight, light and medium vulnerability areas still accounts for a larger proportion. If the negative human interference is reduced and active and effective measures for ecological protection and restoration are taken, the ecosystem will gradually improve and the ecological vulnerability will weaken. On the contrary, if the environmental problems of the ecosystem are not given sufficient attention and the ecosystem continues to be destroyed, the ecologically vulnerable areas will increase.

## Figures and Tables

**Figure 1 ijerph-20-01525-f001:**
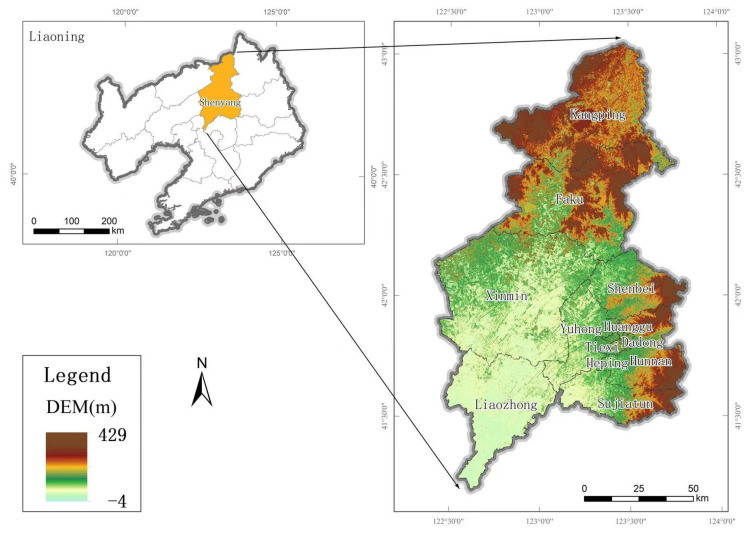
Schematic diagram of Shenyang’s location in Liaoning Province.

**Figure 2 ijerph-20-01525-f002:**
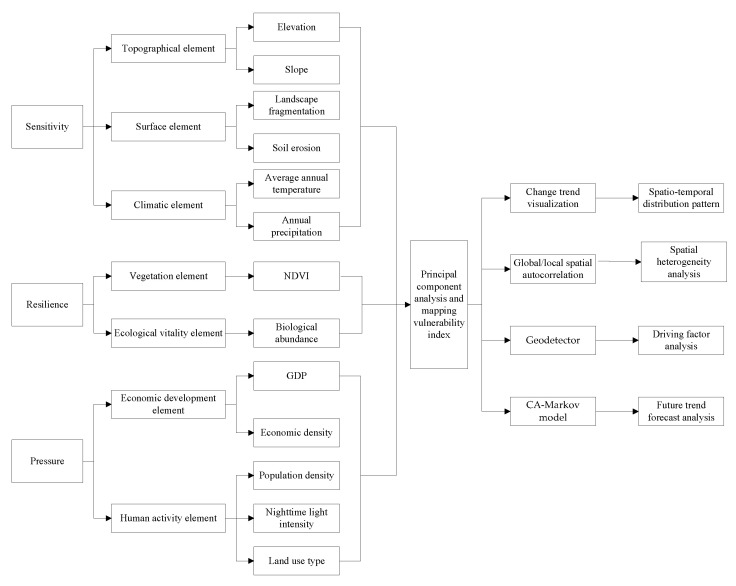
Analysis framework of ecological vulnerability evaluation system.

**Figure 3 ijerph-20-01525-f003:**
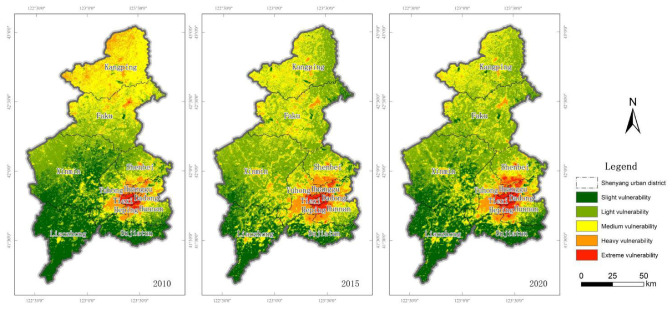
Spatial classification of ecological vulnerability in Shenyang in 2010, 2015 and 2020.

**Figure 4 ijerph-20-01525-f004:**
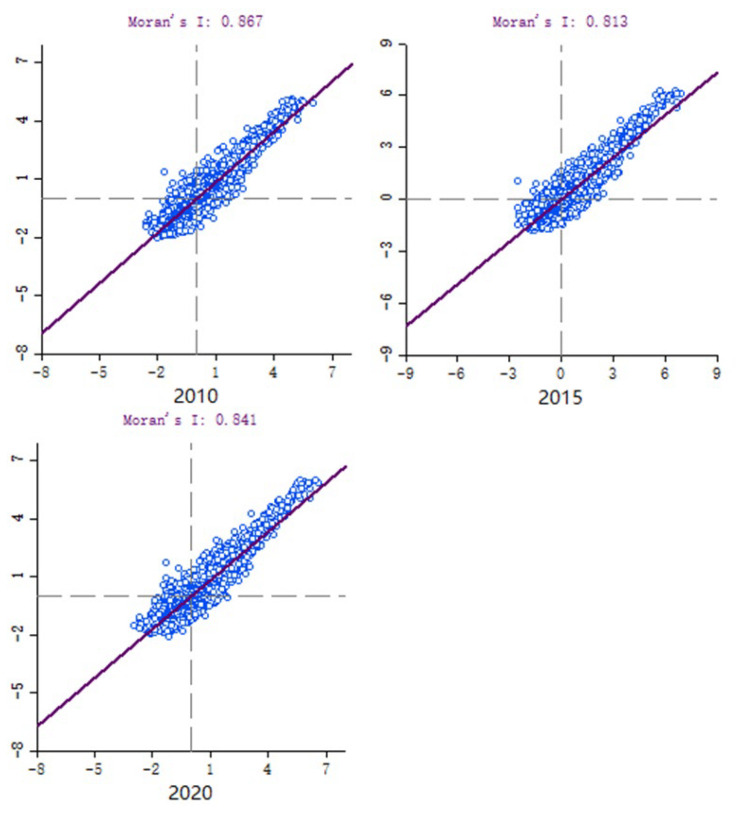
Ecological vulnerability Global Moran′s I scatter plot in Shenyang in 2010, 2015 and 2020.

**Figure 5 ijerph-20-01525-f005:**
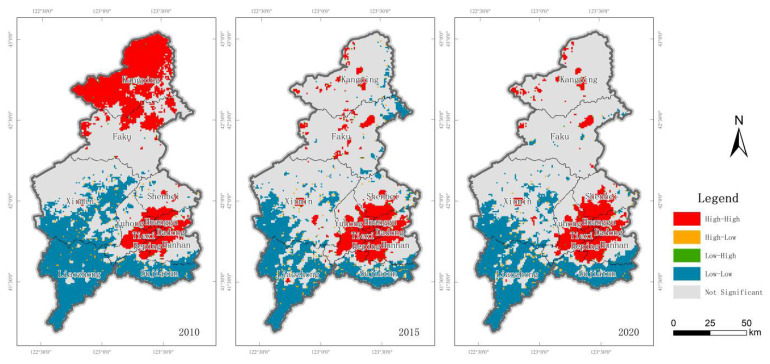
LISA diagram of ecological vulnerability index in Shenyang in 2010, 2015 and 2020.

**Figure 6 ijerph-20-01525-f006:**
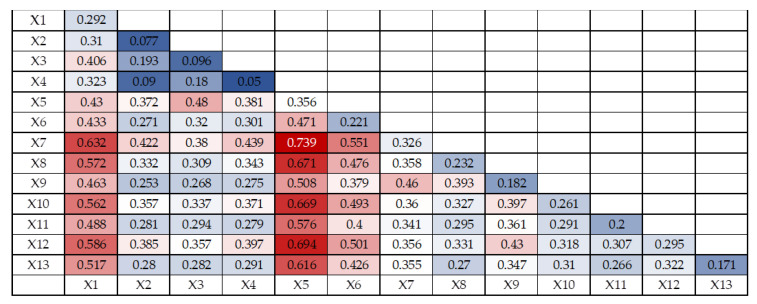
Ecological Vulnerability Interaction Detector Results in Shenyang in 2010.

**Figure 7 ijerph-20-01525-f007:**
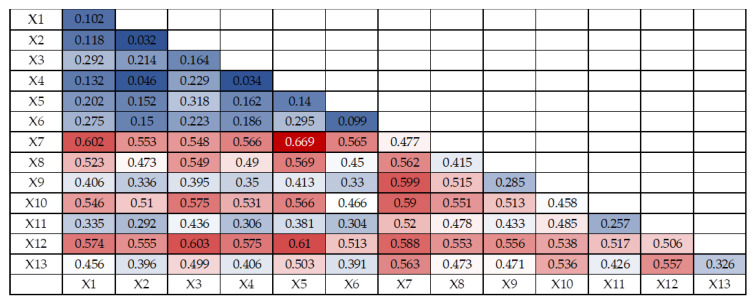
Ecological Vulnerability Interaction Detector Results in Shenyang in 2015.

**Figure 8 ijerph-20-01525-f008:**
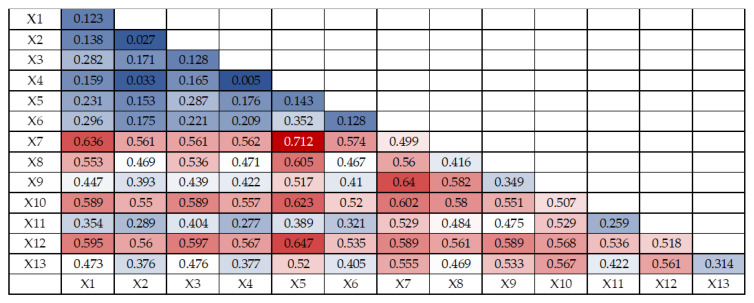
Ecological Vulnerability Interaction Detector Results in Shenyang in 2020.

**Figure 9 ijerph-20-01525-f009:**
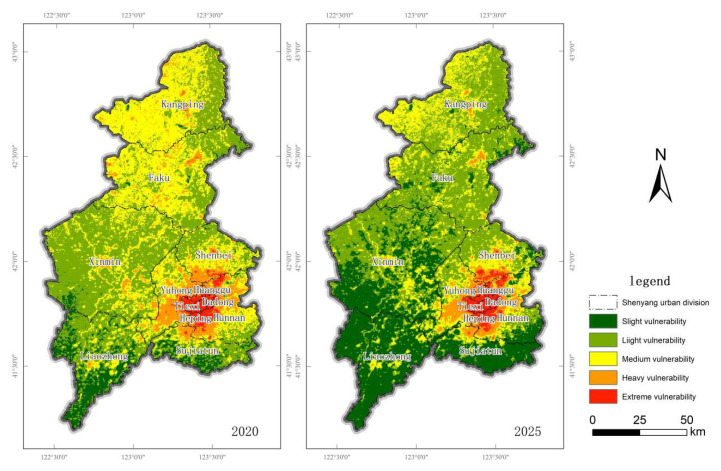
Simulation results of ecological vulnerability in Shenyang in 2020 and 2025.

**Table 1 ijerph-20-01525-t001:** Ecological vulnerability evaluation index system.

Objective		Element	Factor	Factor Properties	Factor Coding
Ecological Vulnerability	Sensitivity	Topographical element	Elevation	Positive	X1
			Slope	Positive	X2
		Surface element	Landscape fragmentation	Positive	X3
			Soil erosion	Positive	X4
		Climatic element	Average annual temperature	Negative	X5
			Annual precipitation	Negative	X6
	Resilience	Vegetation element	NDVI	Negative	X7
		Ecological vitality element	Biological abundance	Negative	X8
	Pressure	Economic development element	Gross national product per capita	Positive	X9
			Economic density	Positive	X10
		Human activity element	Population density	Positive	X11
			Nighttime light intensity	Positive	X12
			Land use type	※	X13

Note: ※Land use type is a type variable, and there is no positive or negative.

**Table 2 ijerph-20-01525-t002:** Land use Assignment in Shenyang.

Land Use	Cropland	Woodland	Grassland	Waters	Construction Land	Bare Land
Value	4	2	3	1	5	6

**Table 3 ijerph-20-01525-t003:** Principal component analysis results.

Year	Principal Component Coefficients	Principal Components					
		PC1	PC2	PC3	PC4	PC5	PC6
2010	Eigenvalue	3.143	1.919	1.866	1.780	1.707	1.053
	Contribution rate	24.175	14.763	14.351	13.690	13.129	8.099
	Cumulative contribution rate	24.175	38.939	53.290	66.980	80.109	88.207
2015	Eigenvalue	2.979	1.894	1.810	1.712	1.706	1.106
	Contribution rate	22.916	14.567	13.923	13.172	13.123	8.506
	Cumulative contribution rate	22.916	37.482	51.406	64.577	77.700	86.206
2020	Eigenvalue	2.989	1.847	1.824	1.814	1.682	1.102
	Contribution rate	22.995	14.206	14.030	13.956	12.936	8.474
	Cumulative contribution rate	22.995	37.201	51.23	65.186	78.122	86.596

**Table 4 ijerph-20-01525-t004:** Ecological vulnerability grading standard of Shenyang.

Vulnerable Area Category	Grade	Ecological Vulnerability Index
Slightly vulnerability	I	0.091~0.226
Lightly vulnerability	II	0.254~0.301
Medium vulnerability	III	0.262~0.303
Heavy vulnerability	IV	0.303~0.368
Extreme vulnerability	V	0.368~0.564

**Table 5 ijerph-20-01525-t005:** Percentage of ecologically vulnerable areas in each district and county of Shenyang.

District and County	Year	Percentage of Area of Ecologically Vulnerability Areas (%)				
		Slightly Vulnerability	Lightly Vulnerability	Medium Vulnerability	Heavy Vulnerability	Extreme Vulnerability
Dadong	2010				43.27%	56.73%
	2015				11.73%	88.27%
	2020				0.05%	99.95%
Faku	2010	0.82%	41.52%	51.15%	6.39%	0.13%
	2015	3.34%	47.46%	43.50%	5.64%	0.06%
	2020	4.03%	62.47%	30.41%	3.07%	0.02%
Heping	2010				2.41%	97.59%
	2015				0.03%	99.97%
	2020				0.14%	99.86%
Huanggu	2010	2.27%	1.61%	1.93%	37.78%	56.42%
	2015	3.07%	1.58%	3.05%	44.65%	47.65%
	2020	1.52%	1.41%	1.85%	26.65%	68.57%
Hunnan	2010	8.76%	29.53%	28.62%	28.58%	4.52%
	2015	3.91%	19.80%	29.29%	34.38%	12.62%
	2020	11.13%	23.59%	21.54%	28.73%	15.01%
Kangping	2010	0.14%	3.42%	71.96%	24.41%	0.07%
	2015	1.26%	41.95%	51.49%	5.24%	0.07%
	2020	1.06%	45.75%	49.12%	4.03%	0.04%
Liaozhong	2010	83.65%	14.38%	1.96%	0.01%	
	2015	56.03%	32.72%	10.30%	0.94%	
	2020	70.57%	22.69%	6.52%	0.22%	
Shenbei	2010	6.35%	54.18%	30.00%	9.46%	0.01%
	2015	3.47%	42.93%	34.88%	17.70%	1.02%
	2020	3.41%	41.11%	34.52%	16.57%	4.38%
Shenhe	2010				0.02%	99.98%
	2015					100.00%
	2020					100.00%
Sujiatun	2010	69.24%	21.41%	5.83%	3.51%	0.01%
	2015	45.01%	32.41%	15.14%	7.17%	0.27%
	2020	62.16%	21.95%	9.86%	5.90%	0.13%
Tiexi	2010				13.17%	86.83%
	2015				8.38%	91.62%
	2020				6.92%	93.08%
Xinmin	2010	33.77%	56.88%	8.81%	0.54%	
	2015	7.83%	67.93%	20.95%	3.27%	0.01%
	2020	21.60%	61.78%	14.96%	1.66%	
Yuhong	2010	12.49%	38.01%	23.99%	23.85%	1.66%
	2015	1.06%	23.92%	31.95%	39.84%	3.23%
	2020	5.08%	29.73%	28.45%	31.69%	5.05%

**Table 6 ijerph-20-01525-t006:** Factor detector results.

Factor	q-Value			Factor	q-Value		
	2010	2015	2020		2010	2015	2020
Elevation	0.292	0.102	0.123	Biological abundance	0.232	0.415	0.416
Slope	0.077	0.032	0.027	Gross national product per capita	0.182	0.285	0.349
Landscape fragmentation	0.096	0.164	0.128	Economic density	0.261	0.458	0.507
Soil erosion	0.050	0.034	0.005	Population density	0.200	0.257	0.259
Average annual temperature	0.356	0.140	0.143	Nighttime light intensity	0.295	0.506	0.518
Annual precipitation	0.221	0.099	0.128	Land use type	0.171	0.326	0.314
NDVI	0.326	0.477	0.499				

**Table 7 ijerph-20-01525-t007:** Changes of ecological vulnerability in Shenyang in 2020 and 2025 (km^2^).

Vulnerability	2020		2025		Change	
	Area	Percentage	Area	Percentage	Area	Percentage
Slightly vulnerability	2621.492	20.38%	3650.532	28.38%	1029.040	8.00%
Lightly vulnerability	5798.601	45.08%	5605.426	43.58%	−193.175	−1.50%
Medium vulnerability	3159.580	24.56%	2539.346	19.74%	−620.233	−4.82%
Heavy vulnerability	931.442	7.24%	697.002	5.42%	−234.440	−1.82%
Extreme vulnerability	351.429	2.73%	370.238	2.88%	18.808	0.15%

**Table 8 ijerph-20-01525-t008:** Transfer of ecologically vulnerable areas in Shenyang in 2020 and 2025 (km2).

2020	2025					
	Slightly Vulnerability	Lightly Vulnerability	Medium Vulnerability	Heavy Vulnerability	Extreme Vulnerability	Decrease
Slightly vulnerability	2605.072	14.495	1.926	—	—	16.420
Lightly vulnerability	1045.461	4699.673	53.467	—	—	1098.928
Medium vulnerability	—	891.258	2265.241	3.081	—	894.339
Heavy vulnerability	—	—	218.713	692.655	20.075	238.787
Extreme vulnerability	—	—	—	1.266	350.163	1.266
Increase	1045.461	905.753	274.106	4.347	20.075	

## Data Availability

Not applicable.

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
