# Peer review of "Spatiotemporal Dynamics of Ecological Vulnerability and Its Influencing Factors in Shenyang City of China: Based on SRP Model"

_ijerph, 2023, doi:10.3390/ijerph20021525_

Round 1
Reviewer 1 Report
This paper assessed the vulnerability of Shengyang with the "sensitivity-resilience-pressure" model to provide some implications for the Territorial Spatial Planning in this city. I agree the vulnerability assessment is an indispensible part to city planning and it deserves deep research. I hope to find some enlightment for the practice or theoretical building of vulnaribility assessment in this paper. However, I didn't find what I expected yet. I suggest the authors to improve their research from two perspective:
Firstly, justify the selection of Shenyang city as a case study with more compelling evidence. Shenyang maybe a city of great significance in China, but it is not well-known for international readers. As far as I know, Shenyang used to be the center of heavy industry in China. The development of heavy industry may expose the natural environment of Shenyang to great vulnerability. The vulnerability assessment of Shenyang can provide some implications for other cities that were featured by heavy industry.
Secondly, introduce the "sensitivity-resilience-pressure" model more clearly. The model is the basis of this research, but the authors only mentioned it with several sentences in the introduction. The logic of this model and the connotation of its three main components, namely the sensitivity, resilience and pressure, should be introduce with more details in the method. A simple presentation of the index is completely not enough.
Reviewer 2 Report
This study aims at exploring the spatial distribution, influencing factors and future trend of ecological fragility in Shenyang, China using spatial autocorrelation analysis and geographic detector model. In general, the study provides a useful framework to analyze ecological vulnerability and is of great significance to guide and balance the relationship between environmental protection and economic development. The reviewer reads through the manuscript and thinks it is appropriate for publication in International Journal of Environmental Research and Public Health.
As regards the manuscript as its stands, the reviewer has the following suggestions which the authors might find helpful.
(1) Why did this article choose 2010, 2015 and 2020 instead of other years or consecutive years? This point needs to be explained.
(2) In the introduction, please add some statements about the comparison and connection between the municipal land space planning and the sustainable development of large and medium-sized cities.
(3) Conclusions and suggestions can be further refined and improved. It is suggested to put forward clear and targeted suggestions in combination with the research results.
(4) Some minor points:
I think the title should be revised to “Study on the Spatial and Temporal Evolution Characteristics of Ecological Vulnerability and Its Influencing Factors in Shenyang, China Based on SRP Model”.
Line 10: replace “the whole Northeast China” with “the whole of Northeast China”
Line 11: replace “important” with “essential”
Line 15: replace “In order to” with “to”
Line 22: delete “by”
Line 59: replace “is” with “are”
Line 69: delete “the”
Line 77: replace “vulnerability” with “vulnerable”
Line 85: delete “in”
Line 108: replace “to practice” with “for practicing”
Line 132: replace “gray” with “grey”
Line 153: delete “as”
Line 165: replace “focus” with “focuses”
Line 393: delete “in”
Line 397: replace “is” with “was”
Line 399: replace “is” with “was”
Line 404: replace “is” with “has been”
Line 425: delete “have”
Line 429: replace “ability” with “abilities”
Line 431: delete “in”
Line 454: replace “decreases” with “decreased”
Line 454: add “,” before “respectively”
Line 460: replace “of” with “from”
Line 478: replace “of” with “in”
Line 497: replace “population” with “Population”
Reviewer 3 Report
Please check Table 1 (line 253)- climate element X5 or X6.
Regarding the introducrion almost all references are from china
In conclusions (2) please underline the reasons or indicators that lead to ecological vulnerability changing eg. from medium to light
Round 2
Reviewer 1 Report
I think my previous comments were not addressed adequately, especially ponit 1. Further revision is needed for this manuscript.
